# A permutation test and spatial cross-validation approach to assess models of interspecific competition between trees

**David Allen** [1]☯*, **Albert Y. Kim** [2]☯

**1** Department of Biology, Middlebury College, Middlebury, VT, United States of America, **2** Statistical and Data Sciences Program, Smith College, Northampton, MA, United States of America

☯ These authors contributed equally to this work.
* dallen@middlebury.edu

**Data Availability Statement:** The data underlying the results presented in the study are available from Deep Blue Data. https://doi.org/10.7302/wx55-kt18.

## Abstract

Measuring species-specific competitive interactions is key to understanding plant communities. Repeat censused large forest dynamics plots offer an ideal setting to measure these interactions by estimating the species-specific competitive effect on neighboring tree growth. Estimating these interaction values can be difficult, however, because the number of them grows with the square of the number of species. Furthermore, confidence in the estimates can be overestimated if any spatial structure of model errors is not considered. Here we measured these interactions in a forest dynamics plot in a transitional oak-hickory forest. We analytically fit Bayesian linear regression models of annual tree radial growth as a function of that tree's species, its size, and its neighboring trees. We then compared these models to test whether the identity of a tree's neighbors matters and if so at what level: based on trait grouping, based on phylogenetic family, or based on species. We used a spatial cross-validation scheme to better estimate model errors while avoiding potentially over-fitting our models. Since our model is analytically solvable we can rapidly evaluate it, which allows our proposed cross-validation scheme to be computationally feasible. We found that the identity of the focal and competitor trees mattered for competitive interactions, but surprisingly, identity mattered at the family rather than species-level.

## Introduction

Competition is a key biotic interaction which structures communities. To better understand the role it plays, we need ways to understand species-specific competitive interactions. Attempts have been made to do this through direct measurement of species-specific competition coefficients, or to generalize competitive interactions based on trait or phylogeny differences between competitors.

Repeat censused forest inventory plots are good places to measure species-specific competition. In such forests trees are identified, mapped, and have their diameter measured at regular time intervals [1, 2]. Diameter growth between censuses of individual trees is modeled as a

**Funding:** This work was supported by the Edwin S. George Reserve Fund and by a USDA McIntyre-Stennis Grant. The funders had no role in study design, data collection and analysis, decision to publish, or preparation of the manuscript.

**Competing interests:** The authors have declared that no competing interests exist.

function of neighborhood tree species identity, size, and distance to focal tree [3, 4]. This gives a way to estimate the species-specific effect of competition on tree growth.

The first question is whether species identity of the competitor matters at all [5], or is competition neutral (sensu [6]). Assuming that the identity of competitors does matter, it can then be hard to measure all of the interaction coefficients, which increase proportionally with the square of species number in the community. Furthermore, some species pairs might not coexist nearby one another enough to get sufficient data to measure their effects on one another. A number of approaches have been taken to deal with large number of parameters to estimate and with missing species pairs.

In low-diversity forests, attempts have been made to measure species-specific competition coefficients of the effect of competition on radial tree growth [7–10]. In a western US, evergreen forest, Das [8] was able to estimate the 16 competition coefficients among the four dominant tree species. Canham et al. [7] measured the competition coefficients between the 14 most common tree species in Northern New England. While they were able to estimate most coefficients, they could not for species-pairs that rarely occurred together.

A hierarchical Bayesian approach can be taken to address this problem [11]. All species-pairs are considered, no data are thrown out, and competition coefficients are "regressed back" to an overall average. The amount of regression decreases with the number of interacting pairs of individuals of that particular species pair. This approach was taken by Tatsumi et al. [9] when looking at 38 tree species in a cool-temperate mixed conifer-broadleaf forest. They found support for the hierarchical model over non-hierarchical version, but because of relatively small sample size, competition coefficients of only a few species pairs had 95% credible intervals that did not overlap with the hyperparameter value.

A final approach is to estimate competition coefficients based on species attributes, rather than measure each one. A common approach is to correlate competition coefficients with either trait or phylogenetic distance between the species [12]. Uriarte et al. [12] found that trait distance did a better job of explaining competitive interactions than phylogenetic distance. Kunstler et al. [13] also found that traits can explain competitive interactions.

For any of these approaches, a method for comparing model performance is needed. Previous studies have used information criteria to compare models [7–9, 12]. Such information criteria are asymptotic approximations for out-of-sample-prediction error [14]. Another commonly used measure of out-of-sample-prediction error is the more easily interpretable (root) mean squared error (RMSE) [15].

Here we present a method to answer two questions from repeat censused forest inventory plots: (1) "Does the species identity of neighboring competitor trees matter?" and (2) "If so, how can you measure species-specific competition coefficients?" For both questions, we analytically derived posterior estimates of coefficients of a linear Bayesian neighborhood competition model. The chief advantage of this analytic approach is that the aforementioned coefficients are much less computationally expensive to estimate than other approaches, such as the Markov chain Monte Carlo based estimates necessitated by other more complicated models [7, 9].

To answer question (1) from the introduction, we use a permutation test based approach. We first fit the model and obtain an observed test statistic assessing the quality of the model fit, in our case the observed RMSE. We then permute the competitor species identities and compute a permuted test statistic a large number of times, thereby empirically generating a "null" distribution of the test statistic. We then compare the observed test statistic to this null distribution to obtain significance measures. The rapid analytic solution is required for this permutation test to run in a reasonable amount of time. We find that identity of the competitor species does matter.

For question (2) from the introduction, we can further ask whether species identity, family identity, or trait values best explains competitive interactions. Here we provide a method to compare spatially cross-validated RMSE's of competing models. The rapid analytic solution is once again required to fit the model repeatedly in the cross-validation scheme. Surprisingly we find that grouping species by family best explains competitive interactions.

## Materials and methods

### Study site and sampling

The Big Woods plot is a 23 ha forest dynamics plot in Pickney, MI (42.462902 N, 84.006093 W). The plot is within a transitional oak-hickory forest. The canopy is dominated by black oak (*Quercus velutina*), northern red oak (*Quercus rubra*), white oak (*Quercus alba*), bitternut hickory (*Carya cordiformis*), and shagbark hickory (*Carya ovata*). However there are relatively few oaks in the mid and understory, instead these strata are dominated by red maple (*Acer rubrum*) and black cherry (*Prunus serotina*). A full list of species found in the plot can be found in S2 Appendix. The elevation in the plot ranges from 270 m to 305 m. Above 275 m the mineral soils are largely Boyer–Oshtemo sandy loam, and below 275 m the soils are mainly histosols dominated by Carlisle and Rifle muck. The rugged topography within the plot is the result of glacial scouring with hills and knobs separated by kettle holes and basins. For more information on the plot see Allen et al. [16].

The original plot was established in 2003 and was only 12 ha. All free-standing woody stems in the plot larger than 3.2 cm diameter at breast height (DBH) were censused, had their DBH measured, mapped, and identified. Between 2007 and 2010 this plot was expanded to 23 ha using the same censusing technique and the original 12 ha were re-censused. In 2014 the entire 23 ha plot was re-censused to determine the diameter growth of each individual, tag individuals recruited into the greater than 3.2 cm DBH size class, and identify which individuals died. For this analysis we considered the average annual diameter growth between the 2007–2010 and 2014 censuses for all stems alive in both censuses ignoring re-sprouts. We defined stems alive in the 2007–2010 census as the competitor stems. DA collected these data with the help of others listed in the Acknowledgements section. The Big Woods plot is located within the Edwin S. George Reserve. Permission to sample in the Reserve was granted by the University of Michigan Department of Ecology and Evolutionary Biology.

The plot is part of the Smithsonian Institution's ForestGEO global network of forest research plots (Smithsonian Institution, Washington DC, USA) [1]. Data from the censuses are available at Allen et al. [17].

### Species grouping methods

We tested which grouping of trees best explained competitive interactions. We grouped species in three ways: (1) by species, (2) by family, and (3) according to traits assumed to correlate with competitive interactions. To form this third grouping, we collected species trait values from the TRY database of plant traits [18]. From this database we picked plant height, wood density, and specific leaf area as our traits of interest because they have been identified as important to tree competition [13]. We clustered these species based on their values for these three traits using the `cluster` package in R [19, 20]. The distance matrix was formed by the euclidean distance between species' three trait values. We then formed a hierarchical clustering of this distance matrix using the `agnes` command. We chose to cut the resulting hierarchical tree into six groups.

## Neighborhood-effect growth model

Let $i = 1, \ldots, n_j$ index all $n_j$ trees of "focal" species group $j$; let $j = 1, \ldots, J$ index all $J$ focal species groups; and let $k = 1, \ldots, K$ index all $K$ "competitor" species groups. We modeled the growth in diameter per year $y_{ij}$ (in centimeters per year) of the $i^{th}$ tree of focal species group $j$ as a linear model $f$ of the following covariates $\vec{x}_{ij}$

$$y_{ij} = f(\vec{x}_{ij}) + \epsilon_{ij} = \beta_{0,j} + \beta_{\text{DBH},j} \cdot \text{DBH}_{ij} + \sum_{k=1}^{K} \lambda_{jk} \cdot \text{BA}_{ijk} + \epsilon_{ij} \tag{1}$$

where $\beta_{0,j}$ is the diameter-independent growth rate for group $j$; $\text{DBH}_{ij}$ is the diameter at breast height (in centimeters) of the focal tree at the earlier census; $\beta_{\text{DBH},j}$ is the amount of the growth rate changed depending on diameter for group $j$; $\text{BA}_{ijk}$ is the sum of the basal area of all trees of competitor species group $k$ within a neighborhood of 7.5 meters of the focal tree; $\lambda_{jk}$ is the change in growth for individuals of group $j$ from nearby competitors of group $k$; and $\epsilon_{ij}$ is a random error term distributed Normal$(0, \sigma^2)$. We chose a distance of 7.5 meters as the competitor neighborhood of a focal tree. Other studies have estimated this distance, we used 7.5 meters as an average of estimated values [3, 5, 7, 11].

For focal trees we considered only those alive in both the 2007–2010 and 2014 censuses. Thus this model only considered the effect of competition on growth, not on mortality. Growth between the two censuses was almost always positive, with the few negative values probably reflecting measurement error.

We considered models where the focal species grouping $j = 1, \ldots, J$ and the competitor species grouping $k = 1, \ldots, K$ reflected the three notions of species grouping introduced earlier: trait group with 6 groups, phylogenetic family with 20 groups, and actual species with 36 groups. While our model specification is flexible enough where our notion of focal tree grouping does not necessarily have to match our notion of competitor tree grouping, for simplicity in this paper we only considered models where both notions match and hence $J = K$.

Furthermore, our models incorporated a specific notion of competition reflecting a particular assumption on the nature of competition between trees: *species grouping-specific effects of competition*. For a given focal species group $j$, all competitor species groups exert different competitive effects. Such models not only assumed competition between trees exists, but also that different focal versus competitor species group pairs have different competitive relationships.

More specifically, using the above three notions of species grouping, we compared three different models for growth $y_{ij}$, each with different numbers of $(\boldsymbol{\beta}_0, \boldsymbol{\beta}_{\text{DBH}}, \boldsymbol{\lambda}, \sigma^2)$ parameters estimated.

1. Trait group with $J = K = 6$, thus $(\boldsymbol{\beta}_0, \boldsymbol{\beta}_{\text{DBH}}, \sigma^2)$ consisted of $6 + 6 + 1 = 13$ parameters. Furthermore, there were $6 \times 6$ unique values of $\lambda_{jk}$, thus $\boldsymbol{\lambda}$ is a $6 \times 6$ matrix, totaling $13 + 6 \times 6 = 49$ parameters.

2. Phylogenetic family with $J = K = 20$, thus $(\boldsymbol{\beta}_0, \boldsymbol{\beta}_{\text{DBH}}, \sigma^2)$ consisted of $20 + 20 + 1 = 41$ parameters. Furthermore, there were $20 \times 20$ unique values of $\lambda_{jk}$, thus $\boldsymbol{\lambda}$ is a $20 \times 20$ matrix, totaling $41 + 20 \times 20 = 441$ parameters.

3. Actual species with $J = K = 36$, thus $(\boldsymbol{\beta}_0, \boldsymbol{\beta}_{\text{DBH}}, \sigma^2)$ consisted of $36 + 36 + 1 = 73$ parameters. Furthermore, there were $36 \times 36$ unique values of $\lambda_{jk}$, thus $\boldsymbol{\lambda}$ is a $36 \times 36$ matrix, totaling $73 + 36 \times 36 = 1369$ parameters.

For each of these three models, all $(\boldsymbol{\beta}_0, \boldsymbol{\beta}_{\text{DBH}}, \boldsymbol{\lambda}, \sigma^2)$ parameters were estimated via Bayesian linear regression. We favored a Bayesian approach since it allowed us to incorporate prior

information about all the parameters in Eq (1) [21]. This in turn served us when particular species grouping pairs were rare, leading to posterior distributions that are more weighted towards the prior distribution than the likelihood [21]. While the linear regression model in Eq (1) is much less complex than the model formulations considered by [7], both the posterior distribution and posterior predictive distribution of all parameters have analytic and closed-form solutions. Thus, we are saved from the computational expense of using methods to approximate all posterior distributions such as Markov chain Monte Carlo [22, 23]. These savings in computational expense were important given the large number of times we fit the model in Eq (1), as we outline in the upcoming sections on the permutation test and spatial cross-validation scheme we used.

We present a brief summary of the closed form solutions to Bayesian linear regression here, leaving fuller detail in S1 Appendix. For simplicity of notation, let $\boldsymbol{\beta}$ represent the parameters $\boldsymbol{\beta}_0, \boldsymbol{\beta}_{\text{DBH}}, \lambda$. The likelihood function $p(\boldsymbol{y}|\boldsymbol{\beta}, \sigma^2)$ of our observed growths resulting from Eq (1) is Multivariate Normal $(X\boldsymbol{\beta}, \sigma^2 I_n)$ where $I_n$ is the $n \times n$ identity matrix. Bayesian linear regression exploits the fact that the Multivariate Normal-inverse-Gamma distribution is a conjugate prior of the Multivariate Normal distribution. So given our Multivariate Normal likelihood $p(\boldsymbol{y}|\boldsymbol{\beta}, \sigma^2)$, by assuming that the joint prior distribution $\pi(\boldsymbol{\beta}, \sigma^2)$ of $\boldsymbol{\beta}, \sigma^2$ is $NIG(\boldsymbol{\mu}_0, V_0, a_0, b_0)$ with the following hyperparameters: 1) a mean vector $\boldsymbol{\mu}_0$ for $\boldsymbol{\beta}$, 2) a shape matrix $V_0$ for $\boldsymbol{\beta}$, 3) shape $a_0 > 0$ for $\sigma^2$, and scale $b_0 > 0$ for $\sigma^2$, the joint posterior distribution $\pi(\boldsymbol{\beta}, \sigma^2|\boldsymbol{y})$ is also $NIG(\boldsymbol{\mu}^*, V^*, a^*, b^*)$ with:

$$\boldsymbol{\mu}^* = (V_0^{-1} + X^T X)^{-1}(V_0^{-1}\boldsymbol{\mu}_0 + X^T\boldsymbol{y}) \tag{2}$$

$$V^* = (V_0^{-1} + X^T X)^{-1} \tag{3}$$

$$a^* = a_0 + \frac{n}{2} \tag{4}$$

$$b^* = b_0 + \frac{1}{2}\left[\boldsymbol{\mu}_0^T V_0^{-1}\boldsymbol{\mu}_0 + \boldsymbol{y}^T\boldsymbol{y} - \boldsymbol{\mu}^{*T} V^{*-1}\boldsymbol{\mu}^*\right] \tag{5}$$

It can be shown that the marginal prior distribution $\pi(\sigma)$ is Inverse-Gamma$(a_0, b_0)$ while $\pi(\boldsymbol{\beta})$ is Multivariate-$t$ with location vector $\boldsymbol{\mu}_0$, shape matrix $\Sigma_0 = \frac{b_0}{a_0} V_0$, and degrees of freedom $v_0 = 2a_0$. Given the aforementioned prior conjugacy, the marginal posterior distributions $\pi(\sigma|\boldsymbol{y})$ is also Inverse-Gamma while $\pi(\boldsymbol{\beta}|\boldsymbol{y})$ is also Multivariate-$t$, both with updated $\boldsymbol{\mu}^*, V^*, a^*, b^*$ hyperparameter values in place of $\boldsymbol{\mu}_0, V_0, a_0, b_0$.

Furthermore, it can also be shown that the posterior predictive distribution $p(\tilde{\boldsymbol{y}}|\boldsymbol{y})$ for a model matrix $\tilde{X}$ corresponding to a new set of observed covariates is Multivariate-$t$ with location vector $\tilde{X}\boldsymbol{\mu}^*$, shape matrix $\frac{b^*}{a^*}(I + \tilde{X}V^*\tilde{X}^T)$, and degrees of freedom $v^* = 2a^*$. The means of the posterior predictive distributions will be used to obtain fitted/predicted values $\hat{y}_{ij}$.

## Permutation test

Recall from the previous section our model assumes *species grouping-specific effects of competition*, whereby for a given focal species group $j$, all competitor species groups exert different competitive effects. We evaluated the validity of this hypothesis with the following hypothesis

test:

$$H_0 : \lambda_{jk} = \lambda_j \text{ for all } k = 1, \ldots, K \quad (6)$$

$$\text{vs.} \quad H_A : \text{at least one } \lambda_{jk} \text{ is different} \quad (7)$$

where the null hypothesis $H_0$ reflects a hypothesis of *no species grouping-specific effects of competition* while the alternative hypothesis $H_A$ reflects a hypothesis of *species grouping-specific effects of competition* of our three models.

We could therefore answer question (1) from the introduction of whether the species identity of neighboring competitor trees matters using a permutation test (also called an "exact test"). We generated the null distribution of a test statistic of interest by randomly permuting the competitor species group labels $k$ for all trees of focal species group $j$ for a large number of iterations. Such permutations of the competitor species group labels (while holding all other variables constant) were permissible under the assumed null hypothesis above. After computing the test statistic for each iteration, we then compared this null distribution to the observed value of the test statistic to obtain measures of statistical significance. Given the large number of permutations and the corresponding large number of model fits this required, having the computationally inexpensive parameter estimates discussed above was all the more important.

Our test statistic was a commonly-used and relatively simple measure of a model's predictive accuracy: the root mean-squared error (RMSE) between all observed values $\boldsymbol{y}$ and all fitted/predicted values $\hat{\boldsymbol{y}}$. Specifically, we compared all observed growths $y_{ij}$ with their corresponding fitted/predicted growths $\hat{y}_{ij}$ obtained from the posterior predictive distributions:

$$\text{RMSE}(\boldsymbol{y}, \hat{\boldsymbol{y}}) = \sqrt{\frac{1}{n} \sum_{j=1}^{J} \sum_{i=1}^{n_j} (y_{ij} - \hat{y}_{ij})^2} \quad (8)$$

## Spatial cross-validation

Among the most common methods for estimating out-of-sample predictive error is cross-validation, whereby independent "training" and "test" sets are created by resampling from the original sample of data. The model $\hat{f}$ is first fit to the training set and then the model's predictive performance is evaluated on the test data [15]. However given the spatial structure of forest census data, there most likely exists spatial-auto-correlation between the individual trees and thus using individual trees as the resampling unit would violate the independence assumption inherent to cross-validation. One must instead resample spatial "blocks" of trees when creating training and test data, thereby preserving within block spatial-auto-correlations. Roberts et al. [24] demonstrated that ignoring such spatial structure can lead to model error estimates that are overly optimistic and thus betray the true performance of any model's predictive ability on new out-of-sample data.

To study the magnitude of this optimism, we report two sets or RMSE's: one where both the model was fit and the RMSE evaluated on the same entire dataset and another where cross-validation was performed. On top of the large number of permutations, given the large number of iterations of model fits cross-validation required, the importance of computationally inexpensive parameter estimates discussed earlier was again critical.

In Fig 1 we display the spatial distribution of the 27,192 trees in the Big Woods study region and illustrate one iteration of the spatial cross-validation algorithm. We superimposed a $100 \times 100$ meter grid onto the study region and then assigned each of the 27,192 trees to one of 23 arbitrarily numbered spatial blocks. In this particular iteration of the algorithm, we first fit

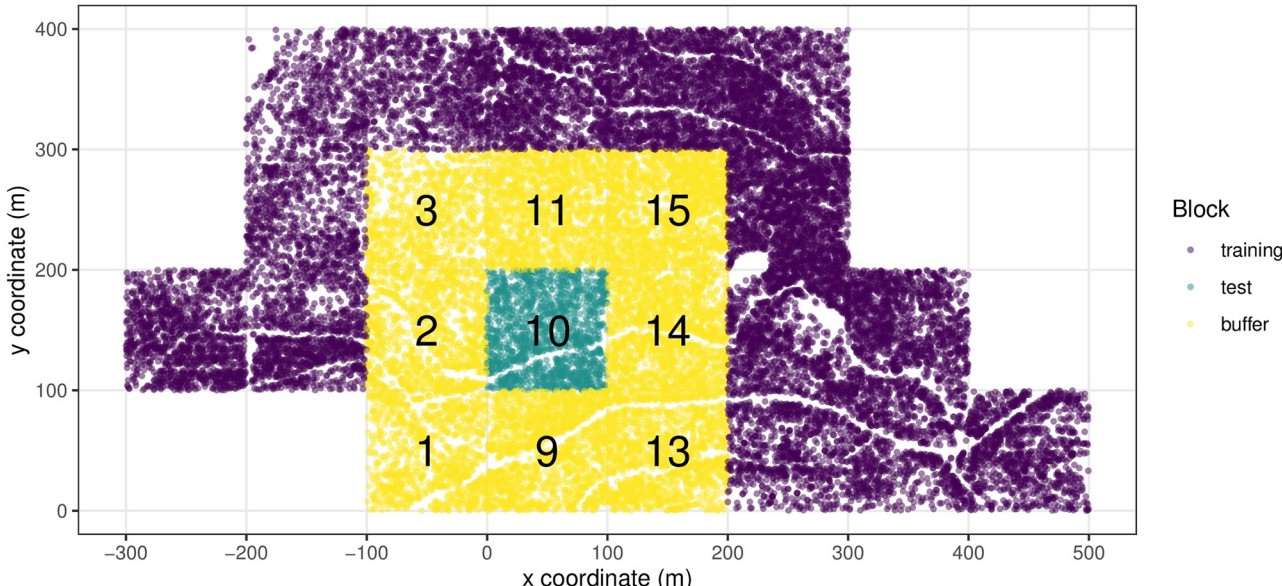

**Fig 1. Big Woods study region with one iteration of spatial cross-validation algorithm displayed.** For this particular iteration of the cross-validation algorithm, we train our models on all 14 unnumbered blocks of trees. We then apply the fitted model to make predictions on block 10. Blocks 1, 2, 3, 9, 11, 13, 14, and 15 act as "buffer" blocks isolating the test block from the training blocks.

our model $\hat{f}$ to all trees in the 14 unnumbered "training" blocks. We then applied the fitted model $\hat{f}$ to all trees in the "test" block labeled 10 to obtain predicted growths $\hat{\boldsymbol{y}}$, which we then compared to observed growths $\boldsymbol{y}$ to obtain the estimated RMSE for this block. The remaining 8 labeled blocks 1, 2, 3, 9, 11, 13, 14, and 15 acted as "buffer" blocks isolating the test block from the training blocks, thereby ensuring that they are spatially independent. We iterated through this process with all 23 blocks acting as the test block once and then averaged the resulting 23 estimated values of the RMSE to obtain a single estimated RMSE of the fitted model $\hat{f}$'s predictive error.

While other implementations of cross-validation exist, in particular implementations that incorporate more principled approaches to determining grid sizes [25], we favored the above blocked approach for its ease of implementation and understanding.

We wrote and fit the model in R and used the `tidyverse` and `ggrdiges` packages [20, 26, 27].

# Results

## Permutation test and cross-validation results

We used a permutation test to evaluate whether the identity of the competitor matters for competitive interactions, and used spatial cross-validation to evaluate which of the three identity groupings (trait, family, or species) yielded the best model. For all three groupings the identity of the competitor did matter; the RMSE with actual competitor identity was less than the RSME when the competitor identity was permuted (Fig 2 compare the dotted horizontal lines to the histograms). To address question (2) from the introduction, which species-grouping does the best job of describing competition, we compare subpanels in Fig 2. The trait-grouping model performed much worse than the two phylogenetic groups models. Without spatial cross-validation the species-level grouping greatly out performs the other two models, but

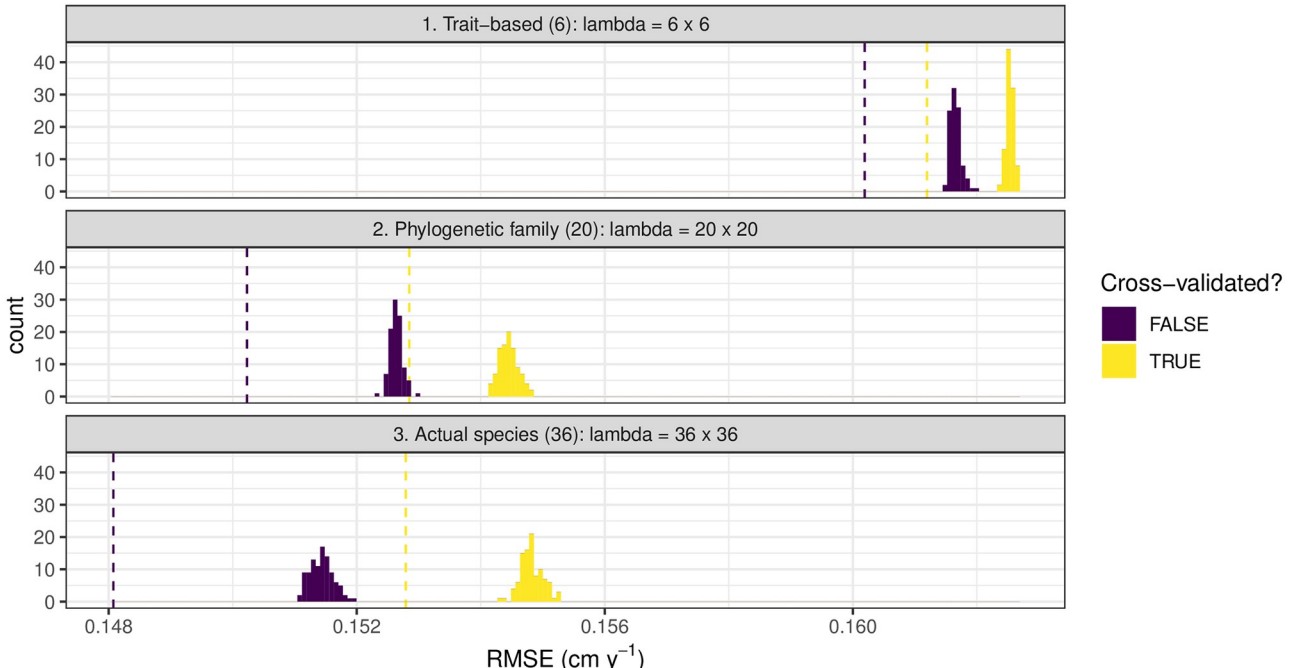

**Fig 2. Comparison of the RMSE for all three species-groupings models.** We calculated the RMSE with and without spatial cross-validation. The dotted lines indicate the RMSE value when the competitor species' identities were not permuted. The histograms indicate the distribution of the RMSE values resulting from 99 permutations of competitor species' identities.

with cross-validation the species and family models perform just as well, suggestive of over-fitting of the species-level model. We will use the family-level model for the remainder of the paper, as it performs just as well as the species-level model but with many fewer parameters.

## Posterior distributions of parameters of best model

We found that the family-level model performed nearly as well as the species-level one (compare the dashed yellow lines in panels 2 and 3 of Fig 2). It did so with many fewer parameters, so here we will report the posterior distributions of relevant ($\boldsymbol{\beta}_0$, $\boldsymbol{\beta}_{\mathrm{DBH}}$, $\lambda$) parameters for the family-level model. In Fig 3, we plot the posterior distribution of all $\beta_{0,j}$ baselines for all families, $j$. These values range from 0.05 to 0.4 cm y$^{-1}$. We generally have better estimates of parameters for families with larger sample sizes. In Fig 4, we plot the posterior distribution of all $\beta_{\mathrm{DBH},j}$; this is the effect of tree DBH on growth rate. For most families these values are between 0 and 0.05. These positive values indicate that larger trees grow faster. For the few families with estimates of negative $\beta_{\mathrm{DBH},j}$, smaller trees grow faster.

In Fig 5, we plot the posterior distribution of all $\lambda$ relating to inter-family competition. For clarity this figure shows the $\lambda$ values for the eight families with at least 200 individuals in the plot (for the full 20-by-20 $\lambda$ matrix see S1 Fig). Fig 5 shows differences in family-level *effect* and *response* to competition. Some families, such as Fagaceae and Juglandaceae, have a strong negative effect on the growth of all other families. In other words, for focal trees of nearly all families having many Fagaceae and Juglandaceae (oaks and hickories) neighbors is associated with slower growth. Other families, such as Rosaceace, have a positive effect on the growth of most other families. In other words, for focal trees of nearly all families having many Rosaceace neighbors is associated with faster growth. Generally there is a strong negative intra-family effect even for families which have little negative or even a positive effect on other families, for

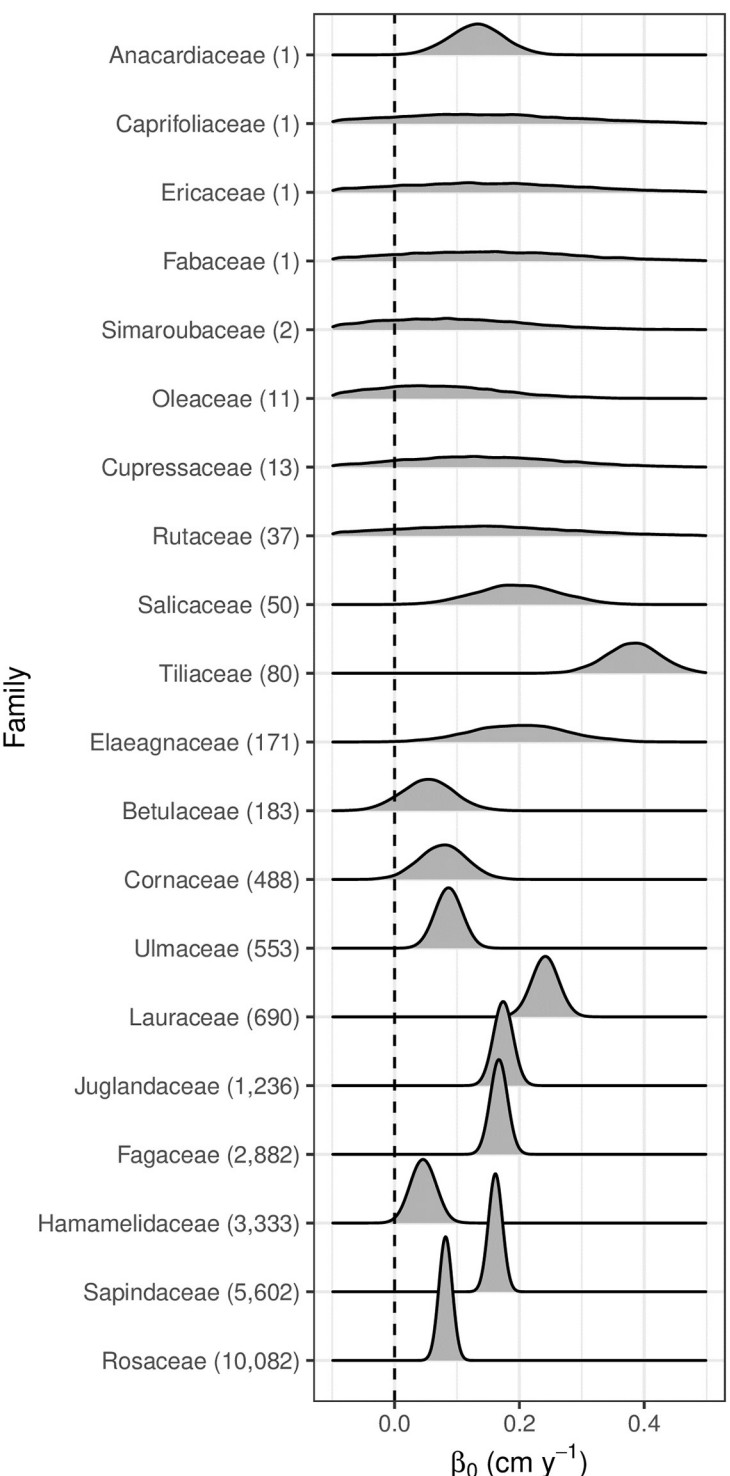

**Fig 3. Posterior distribution of $\beta_{0,j}$ for the family-level model.** These distributions display the estimated baseline (diameter-independent) growth (cm $y^{-1}$) for each family.

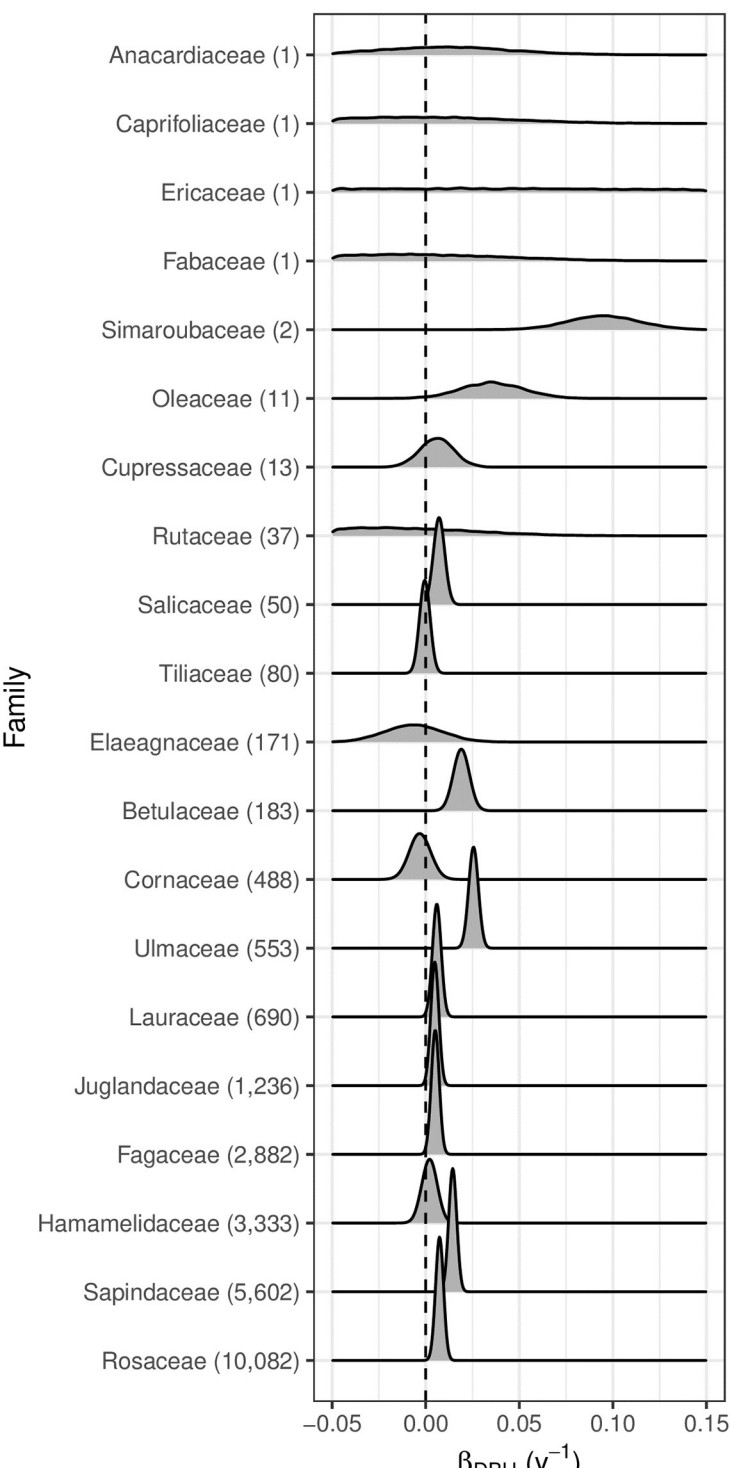

**Fig 4. Posterior distribution of $\beta_{\text{DBH},j}$ for the family-level model.** These distributions display the estimated change in annual growth (cm y$^{-1}$) per cm of DBH for each family. Positive values indicate that larger individuals grow faster, while negative values indicate that larger individuals growth slower.

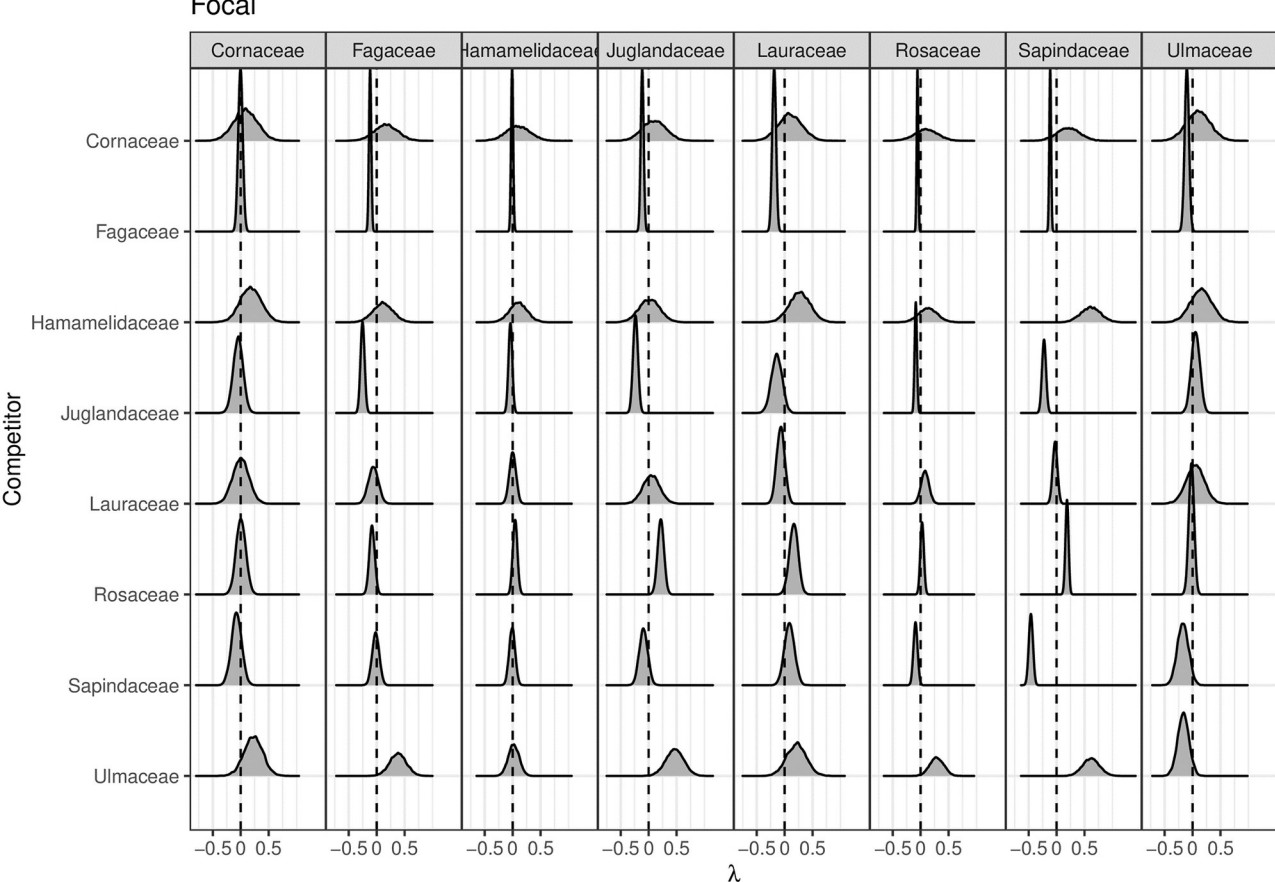

**Fig 5. Posterior distribution of λ family-specific competition coefficients.** Read across rows for that family's competitive *effect* on other families and down columns for that family's *response* to competition from other families. Positive values of λ indicate that trees of the focal group tend to grow faster if they have more neighbors of that competitor group, while negative values of λ indicate that trees of the focal group then to grow slower. For example, almost all groups tend to have slower growth in the presence of more Fagaceae neighbors, but tend to have faster growth in the presence of more Ulmaceae neighbors. Here we display just the 8 families for which there are at least 200 individuals in the plot. For a high resolution version of this image for the full 20-by-20 lambda matrix see S1 Fig. A full list of all species and families in the plot can be found in S2 Appendix. See S2 Fig for a phylogeny of these families and S3 Fig for counts of focal and competitor family pairs.

example Ulmaceae and Sapindaceae. In other words, most individuals tend to grow slower when they have more neighbors of the same family.

Much like the posterior distributions of $\boldsymbol{\beta}_0$ and $\boldsymbol{\beta}_{\mathrm{DBH}}$ shown in Figs 3 and 4, we generally have more precise posterior distributions for the values of λ for focal and competitor family pairs with a larger sample size; for reference S3 Fig shows the counts of such pairs for the eight families in Fig 5.

## Spatial patterns in residuals

We calculated the residuals for all individuals based on the family-level model and plotted them spatially across the plot (Fig 6). There is a clear spatial pattern to these residuals, with spatial patches of trees growing faster than predicted by the model, for example around (0, 150). This is a relatively wet portion of the plot, so soil moisture may be an important factor not considered in the model. In other areas the trees are growing slower than predicted by the model, for example around (450, 50). This suggests that some spatially correlated factor beyond tree species, diameter, or competitors is important to determining tree growth.

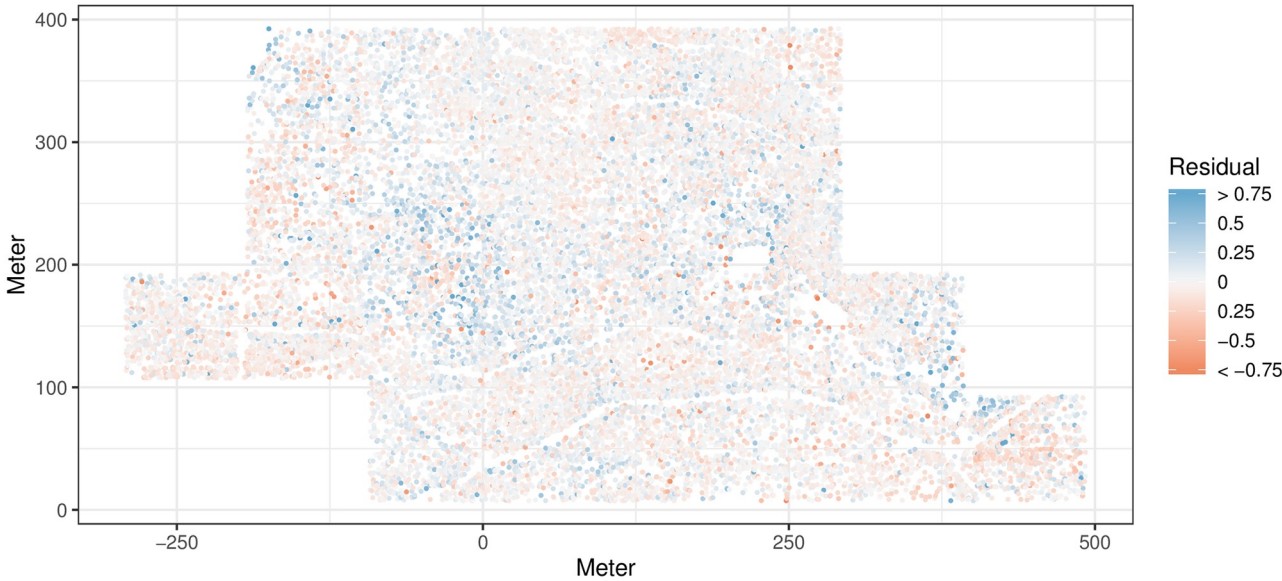

**Fig 6. Spatial pattern of residuals for the family-level model.** This shows the actual growth $y$ of each tree minus its predicted growth $\hat{y}$. Blue individuals grew faster than expected by the model and red individuals grew slower. The residuals show clear spatial patterning.

## Discussion

Here we tested whether species identity matters for competitive interactions among forest trees. We found a strong signal that identity matters for both the effect focal trees feel from competition and the effect competitors exerts on a focal tree. Models which included competitor identity outperformed those with randomized identity. This importance of competitor identity has been observed before [3, 8, 13]. Surprisingly, we found that the model which predicted the effect of competition best was at the family-level rather than species or trait grouping-level. Previous work has found that species traits perform well at predicting competitive ability [12, 13]. We suspect that this family-level grouping for competition is not a general phenomenon, but a consequence of the particular community in this forest. The forest is undergoing rapid successional change as the oak-hickory overstory is being replaced by more mesic, shade-tolerant species [16, 28]. Oaks and hickories here fill a specific functional role, and we suspect the family-level grouping fit best as a result since it groups many species into these two families.

Here we used a Bayesian framework to measure the full $\lambda$ matrix of family-level competitive interactions. This matrix showed clear family-level differences in both the *effect* and *response* to competition. Interestingly, some families had a positive effect on the growth of neighbors, such as Ulmaceae. This is probably not a direct positive effect, but a reflection of the spatial pattern of Ulmaceae individuals. These individuals could inhabit more productive soils, thus nearby individuals grow faster. This illustrates a limitation of neighborhood-based methods for measuring competition [4]. The best way to address this and truly measure competitive interactions would be with a manipulative experiment. This would be largely infeasible for competing forest trees. However, one way to potentially address this issue still in an observational, neighborhood competitor framework would be to control for soil or other covariates potentially important to growth in the model. That way those spatially varying covariates could be included and thus the signal from competition could potentially be more clear.

One clear pattern seen is that oaks and hickories are particularly strong competitors. Their neighbors had consistently lower growth rates. This is interesting in light of the successional change going on in the forest. Allen et al. [28] show that canopy oaks are rapidly being replaced by red maples and black cherries. Even though we found here that oaks had very strong negative effect on neighbors' growth, that is not enough for them to maintain canopy dominance [28].

We fit Bayesian linear regression models of the growth of a focal tree as a function of its species, its size, and the basal area of its competitors. While more complicated models of growth exist than the one proposed in Eq (1), this particular model has the benefit of having closed-form analytic solutions and thus we can avoid computationally expensive methods for posterior estimation such as Markov chain Monte Carlo. This is of particular importance given the large number of times we fit models when performing our permutation test as well as our spatial cross-validation algorithm.

We highlight the importance of cross-validation. It initially appeared in our non cross-validated error estimates in Fig 2 that the species-level model out performed the family-level model. However, when using cross-validation to generate our estimates of model error, both models roughly performed the same. This was due to the over-fitting induced by the large number of parameters of the more complex species-level model. Furthermore, had we not incorporated the inherent spatial structure of our data to our cross-validation algorithm, our model error estimates would have been overly-optimistic. This is a point that been demonstrated in other ecological settings [24, 29, 30].

Here we provide a flexible method to estimate species-specific competitive effects between forest trees. This method could be used for other ForestGEO plots with two or more censuses or with USFS FIA data. Comparisons across forest plots would be particularly powerful to assess whether species-specific interactions are general or site-specific. In the future we hope to produce an R package that includes functions to perform the method presented here.

## Supporting information

**S1 Fig. Posterior distribution of λ values, family-specific competition coefficients.** Read across rows for that family's competitive *effect* on other families and down columns for how a family *responses* to the competition of other families.
(PDF)

**S2 Fig. Phylogenetic relationship of families.** The phylogenetic relationship of families pulled from the Open Tree of Life [31] using the R package `rotl` [32]. A) The phylogeny for the most common families, this corresponds to the families shown in Fig 5. B) The phylogeny for all families in the plot, this corresponds to families shown in S1 Fig.
(PDF)

**S3 Fig. Counts of focal and competitor family pairs.** The total width of each horizontal bar represents the total number of neighbors (or competitors) of focal trees of a particular family in the study region. Within each bar, the width of each color represents the total number of competitor trees of a particular family within a neighborhood of 7.5 meters of trees of the focal family. For clarity this figure shows counts for the eight families with at least 200 individuals in the plot. This figure provides sample sizes for Fig 5.
(PDF)

**S1 Appendix. Closed-form solutions for Bayesian linear regression.**
(PDF)

**S2 Appendix. Species list.**
(PDF)

## Acknowledgments

We thank the individuals who helped census the plot: Omodele Ajagbe, Bob Barretto, Hillary Butterworth, Richard Byler, Vera Chan, Ben Crotte, David Hudson, Lindsay Ford, Katie Gallagher, Jasmine Gramling, Kate Heflick, Rodica Kocur, Carley Kratz, Rachael Lacey, Isaac Levine, Kathleen Parks, Andrew Phillips, Jayna Sames, Margot Sands, John Schroeder, Leah Spaulding, Ethan Strayer, Jordan Trejo, Justin Waraniak, Padhma Venkitapathy, Olivia Velzy, and Ash Zemenick. We also thank Ryan Giordano and Jonathan Che for their help with the statistical methodology. Chris Dick, Ivette Perfecto, and John Vandermeer, with D.A., are PIs of the Big Woods plot. We thank Stuart Davies and Smithsonian staff for guidance in establishing a ForestGEO plot within the E.S. George Reserve.

## Author Contributions

**Conceptualization:** David Allen, Albert Y. Kim.

**Funding acquisition:** David Allen.

**Methodology:** David Allen, Albert Y. Kim.

**Visualization:** David Allen, Albert Y. Kim.

**Writing – original draft:** David Allen, Albert Y. Kim.

**Writing – review & editing:** David Allen, Albert Y. Kim.

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
