## [Decision Letter · Decision Letter 0]

9 Dec 2019

PONE-D-19-24831

A permutation test and spatial cross-validation approach to assess models of interspecies competition between trees

PLOS ONE

Dear Dr Allen,

Thank you for submitting your manuscript to PLOS ONE. After careful consideration, we feel that it has merit but does not fully meet PLOS ONE’s publication criteria as it currently stands. Therefore, we invite you to submit a revised version of the manuscript that addresses the points raised during the review process.

We would appreciate receiving your revised manuscript by Jan 23 2020 11:59PM. To enhance the reproducibility of your results, we recommend that if applicable you deposit your laboratory protocols in protocols.io, where a protocol can be assigned its own identifier (DOI) such that it can be cited independently in the future. For instructions see: http://journals.plos.org/plosone/s/submission-guidelines#loc-laboratory-protocols

We look forward to receiving your revised manuscript.

Kind regards,

Jian Yang

Academic Editor

PLOS ONE

Journal Requirements:

1)

2) Please upload a copy of Figure 7, to which you refer in your text on page 9. If the figure is no longer to be included as part of the submission please remove all reference to it within the text.

Reviewers' comments:

Reviewer's Responses to Questions

**Comments to the Author**

1. Is the manuscript technically sound, and do the data support the conclusions?

Reviewer #1: Yes

Reviewer #2: Yes

2. Has the statistical analysis been performed appropriately and rigorously? 

Reviewer #1: Yes

Reviewer #2: Yes

3. Have the authors made all data underlying the findings in their manuscript fully available?

Reviewer #1: No

Reviewer #2: Yes

4. Is the manuscript presented in an intelligible fashion and written in standard English?

Reviewer #1: Yes

Reviewer #2: Yes

5. Review Comments to the Author

Reviewer #1: In community ecology, one of the most challenging questions in is to clarify the mechanisms of community assembly. Two theories have been widely implemented to explain species coexistence, namely niche theory and neutral theory. The most common modern metaphor of niche theory describes sub pool of species passing through environmental filters and biotic interactions (mainly competition) and then form local communities. So measuring species-specific competitive interactions is key to understanding how communities are assembled. The interspecific competitive interactions in regulating natural plant communities has been investigated in a multitude of studies that have used different methods for answering the question. Well, most studies related to species competition are manipulated experiments where density and/or the proportion of different species are varied and the biomass or fecundity of the competing species are measured. Such competition experiments are often conducted in artificial environmental conditions with a limited number of individuals in small plots. However, there has been an increasing awareness that the interspecific interactions critically depend on the abiotic and biotic setting, it is now more common to conduct ecological manipulation experiments in natural plant communities, where the density of either the neighbors (removal experiments) or the target species has been manipulated. This manuscript provides a method to measure interspecies competition between trees using repeat censured forest inventory plots’ data. Overall, I think this study is easy to read and is also timely and important within the field. However the manuscript still needs reliable English editing, especially English tense. Additionally, there are some specific suggestions as following:

Comment 1:

Line 26: “take” or “taken”?

Comment 2:

Line 38: “find” or “found”?

Comment 3:

The English tense problems existed in the Materials and methods section. Please confirm to keep the tenses consistent.

Comment 4:

Please describe the meanings or definitions of β0,j , βdbh,j and λj,k when present the Neighborhood-effect growth model.

Comment 5:

From the legend of Figure 3, I got that the β0,j represented the estimated baseline growth per year. Furthermore, I also got that the βdbh,j represented the estimated increase in annual growth (cm) per DBH (cm) from the legend of Figure 4. In my opinion, it would be better to change the unit in Figure 3 to be (cm/year or cm·y-1) . And what about the unit for βdbh,j ?

Comment 6:

Indeed, the Figure 3 and Figure 4 represented the tree growth with positive values. Whether the negative values represented negative growth due to the trees’ mortality induced by competition, disturbance (e.g. insect, wind) or senescence? Further, how to distinguish these reasons induced tree mortality?

Comment 7:

The Figure 5 displayed the inter-family and intra-family competitive coefficients (λ). Could the authors provide the detailed species and family information in the supporting information?

Comment 8:

Whether the authors could presented the phylogenetic relationships among the species occurred in the field with phylogenetic tree. If the phylogenetic tree could be provided, the authors could distinguish the families with different colors. It would be useful for authors to understand the results shown in Figure 5 from the aspect of phylogeny

Comment 9: line 259-260

The result said that “There is clear spatial patterning to these residuals, with clusters of trees growing faster than the model predicts and other clusters growing slower”. I failed to understand this result shown in Figure 6. How the result shown that the clusters of trees growing faster than the model predicts and other clusters growing slower? Furthermore, what the other clusters represented for?

Comment 10:

The figure 7 was missed in the manuscript.

Comment 11:

Line 268: “found” or “find”?

Comment 12:

Change “illustrations” into “illustrates”.

Comment 13:

In the discussion section, the authors stated a limitation of neighborhood-based methods for measuring competition. So how to tackle this limitation?

Reviewer #2: I enjoyed reviewing this manuscript. It is well written, concise and deals with an important topic in forest sciences. Please find attached file for especific comments, questions, and suggested edits, which I ask authors to address.

6. PLOS authors have the option to publish the peer review history of their article (what does this mean?). If published, this will include your full peer review and any attached files.

Reviewer #1: No

Reviewer #2: No

---

## [Author Response · Author response to Decision Letter 0]

6 Jan 2020

Dear PLOS One Editor:

Thank you for the opportunity to revise and resubmit our manuscript (PONE- D-19-24831). We also thank the reviewers for their thoughtful comments. We have addressed these comments and think that the manuscript is much stronger as a result.

• The data used in this manuscript are now available on the University of Michigan data repository Deep Blue Data. They have a DOI which is linked in the article. https://doi.org/10.7302/wx55-kt18

• We have addressed the grammar and wording suggests of the reviewers. This includes making sure that the manuscript is written in the past tense.

• There was no Figure 7, this was a mistake in the latex code which re- ferred to a figure that didn’t exist. We have removed this reference to the nonexistent figure.

• As suggested by reviewer 1, we have added a species list (Appendix 2) and phylogenetic tree of the families in the plot (Figure S2).

• We have explicitly defined the parameters — β0,j , βDBH,j , λj,k — when we introduce them in the methods section. We made interpretation of them clearer by explaining the fit values in figure legends and the results section. We clearly stated the unit of each parameter (e.g, β0,j is in cm y-1). For λj,k we discussed what it means for this parameter to be positive versus negative.

• As suggested by both reviewers, we make more clear what we mean by the spatial pattern of residuals in Figure 6.

• As suggested by reviewer 1, we have added some discussion on how to overcome the limitation of neighborhood-based approaches for measuring competition.

• Reviewer 2 suggested that we remove the last two paragraphs from the introduction. The second to last paragraph summarized our methods and the last paragraph summarized our results. We feel that this summary of methods and results in the introduction helps the reader understand the manuscript as a whole. By giving a summary of the methods and results the reader is better able to understand as they go through the manuscript. We prefer including this end-of-introduction summary, but if the reviewers and editor feels strongly that it should be removed we will.

• Reviewer 2 wanted a link to Allen et al. (2019) (line 80 of the revised manuscript). That article is currently in revisions. We think that it will

December 26, 2019

be published by the time this current manuscript is published, so can be updated with final citation. If not we will provide some other access to the manuscript draft. We are happy to share a copy with the reviewer before that if requested.

• We have provided justification for our use of 7.5 m as the distance for the competitive neighborhood. See lines 115–116 of the revised manuscript.

• We have included the sample sizes in Figures 3 and 4. Furthermore, we have included an additional figure of the sample sizes of the focal and competitor families pairs (Figure S3); this will facilitate the interpretation of Figure 5.

Thanks again for the opportunity to resubmit this manuscript. If you have any questions, do not hesitate to ask me.

---

## [Editor Report · Decision Letter 1]

19 Feb 2020

A permutation test and spatial cross-validation approach to assess models of interspecific competition between trees

PONE-D-19-24831R1

Dear Dr. Allen,

We are pleased to inform you that your manuscript has been judged scientifically suitable for publication and will be formally accepted for publication once it complies with all outstanding technical requirements.

With kind regards,

Jian Yang

Academic Editor

PLOS ONE

Additional Editor Comments:

L114: Missing "of" between the word "amount" and "the".

---

## [Editor Report · Acceptance letter]

24 Feb 2020

PONE-D-19-24831R1 

A permutation test and spatial cross-validation approach to assess models of interspecific competition between trees 

Dear Dr. Allen:

I am pleased to inform you that your manuscript has been deemed suitable for publication in PLOS ONE. Congratulations! Your manuscript is now with our production department. 

With kind regards,

on behalf of

Dr. Jian Yang 

Academic Editor

PLOS ONE